# MicroRNAs as New Regulators of Neutrophil Extracellular Trap Formation

**DOI:** 10.3390/ijms22042116

**Published:** 2021-02-20

**Authors:** Sonia Águila, Ascensión M. de los Reyes-García, María P. Fernández-Pérez, Laura Reguilón-Gallego, Laura Zapata-Martínez, Inmaculada Ruiz-Lorente, Vicente Vicente, Rocío González-Conejero, Constantino Martínez

**Affiliations:** Department of Hematology and Medical Oncology, Morales Meseguer University Hospital, Centro Regional de Hemodonación, Universidad de Murcia, IMIB, C/Ronda de Garay S/N, 30003 Murcia, Spain; sonia.aguila@um.es (S.Á.); sregapa@gmail.com (A.M.d.l.R.-G.); mpfernandezperez@gmail.com (M.P.F.-P.); laurareguilongallego@gmail.com (L.R.-G.); laurazap97@gmail.com (L.Z.-M.); irl_98@hotmail.com (I.R.-L.); vicente.vicente@carm.es (V.V.)

**Keywords:** microRNAs, neutrophil extracellular traps, neutrophils, cardiovascular diseases

## Abstract

Neutrophil extracellular traps (NETs) are formed after neutrophils expelled their chromatin content in order to primarily capture and eliminate pathogens. However, given their characteristics due in part to DNA and different granular proteins, NETs may induce a procoagulant response linking inflammation and thrombosis. Unraveling NET formation molecular mechanisms as well as the intracellular elements that regulate them is relevant not only for basic knowledge but also to design diagnostic and therapeutic tools that may prevent their deleterious effects observed in several inflammatory pathologies (e.g., cardiovascular and autoimmune diseases, cancer). Among the potential elements involved in NET formation, several studies have investigated the role of microRNAs (miRNAs) as important regulators of this process. miRNAs are small non-coding RNAs that have been involved in the control of almost all physiological processes in animals and plants and that are associated with the development of several pathologies. In this review, we give an overview of the actual knowledge on NETs and their implication in pathology with a special focus in cardiovascular diseases. We also give a brief overview on miRNA biology to later focus on the different miRNAs implicated in NET formation and the perspectives opened by the presented data.

## 1. Introduction

The recent characterization of neutrophil extracellular traps (NETs) as important physiological immune factors with potential pathological effects has boosted the research on the mechanisms implicated in their formation. To date, many efforts have still to be performed in order to unravel pathways implicated in NET formation. Indeed, this process is complex due in part to the fact that many players are probably still missing. Thus, searching for new elements that may regulate NET formation is crucial to understand this process and to therapeutically fight against it. Among these unknown factors, microRNAs (miRNAs) that have been recognized as new components involved in an elevated number of pathophysiological processes such as cardiovascular diseases may be of relevance. In this review, we will give an overview of the actual knowledge on NETs and their implication in pathology focusing in cardiovascular diseases and a brief glimpse on miRNA biology to later focus on the different miRNAs that have been associated in NET formation and the perspectives opened by the presented data. 

## 2. Neutrophil Extracellular Traps

### 2.1. NET Formation

Neutrophils play a key role in the innate immune system where they represent the major leukocyte population in humans. After an infection or tissue damage, neutrophils are the first leukocytes that are recruited to the inflammatory site [1]. Indeed, it was thought for a long time that phagocytosis was the main neutrophil effector mechanism [2]. In 2004, Brinkmann et al. published a landmark study where they characterized a new mechanism, later named NETosis that is different from necrosis and apoptosis [3], where neutrophils are able to eliminate bacteria during an infectious process by emptying their nuclear content into the extracellular space [4]. NETs are structures composed of nuclear chromatin and associated with nuclear histones with a high cytotoxic function, as well as cytoplasmic and granular antimicrobial proteins able to permeabilize cells [4]. The main function of NETs is to trap and eliminate pathogens avoiding their spread and allowing the concentration of antimicrobial elements at the infection site [5]. In vitro NET formation was first described after activating neutrophils with phorbol-12-myristate-13-acetate (PMA) [4]. This activation was shown to be NADPH oxidase-dependent process via protein kinase C and Raf-MEK-ERK with a concomitant calcium influx [6] finally producing reactive oxygen species (ROS) (Figure 1). This pathway activates a relevant protein, protein-arginine deiminase IV (PAD4), that converts arginine, a positively charged aminoacid, into citrulline within histones and causing chromatin decondensation due to the disruption of the tight association linking histones with DNA [7]. On the other hand, ROS allow the translocation from cytoplasmic azurophilic granules to the nucleus of neutrophil elastase (NE), myeloperoxidase (MPO) and other related enzymes [2]. This translocation further triggers chromatin decondensation and NET formation through histone cleavage [2]. Finally, these processes lead to the loss of nuclear and plasmatic membranes integrity by mechanisms that are still largely unknown (for complete review on NET formation mechanisms refer to Thiam et al. [8]). However, NET formation not always requires cell death. Several scientific evidences point towards the existence of a parallel NET formation process with non-lytic characteristics known as vital NET formation that seems independent of NADPH oxidase [9] (Figure 1). The mechanisms involved in vital NET formation are still elusive. Both vital NET formation and lytic NETosis share some similarities concerning the molecular pathways (NE translocation, PAD4 activation) although the nature of the stimulus [activation through toll like receptors (TLRs) or platelets], as well as the time required for NET to form are different [9]. Importantly, neutrophils undergoing non-lytic NET formation are still able to perform some important functions, such as phagocytosis [10]. 

One of the characteristics of miRNAs that multiplies their regulatory capacity is that they can be extracellularly secreted, packaged in vesicles, exosomes, associated with HDL or bound to proteins such as Argonaute 2 (AGO2) [11,12,13]. In this sense Linhares-Lacerda et al. have described NETs as a new vehicle for miRNAs [14]. Thus, authors showed that miR-142-3p is transferred from NETs to macrophages where it down regulates protein kinase C alpha (PKCα) thus neutralizing the excessive production of tumor necrosis factor alpha (TNFα). Moreover, they described that the expression levels of miRNAs were different depending on how NETosis was induced [14]. These findings would have relevant consequences in a pathophysiological context since the profile of extracellular miRNAs could be different if vital or lytic NETosis is induced. These interesting results need to be further explored to determine, among other issues, whether released miRNAs upon NETosis are fueling neutrophil’s activation or on the contrary, if the components of NETs are driving specific miRNA release by paracrine or exocrine pathways. 

### 2.2. NETs and Pathology

NET formation is designed to restrain infection but given the nature of its components, NETs can promote severe tissue damages to the host. Indeed, the involvement of NETosis in non-infectious diseases such as cancer [15], venous thromboembolism [16], autoimmune diseases [17], or atherosclerosis [18] has been clearly demonstrated [19]. In particular, Engelmann and Massberg established the concept of immunothrombosis that is based on the ability of NETs to induce a procoagulant response that results in thrombus formation [20]. Various components of NETs contribute to the activation of coagulation and the formation of thrombus [21,22]: (i) DNA, whose negative charges activate Factor XII; (ii) histones H3 and H4 are able to activate platelets; (iii) NE degrades two important anticoagulants: tissue factor pathway inhibitor (TFPI) and antithrombin; (iv) cathepsin G blocks TFPI and activates platelets; (v) tissue factor (TF), a central player of the coagulation cascade that activates thrombin generation with the subsequent platelet activation; and (vi) miRNAs that can increase NET formation as described below in the present review. The inclusion of NETosis as a pathophysiological element of cardiovascular disease (CVD) allows the consideration of new elements as useful tools in the diagnosis and/or therapy of these pathologies. Recent research has shown that inflammation, key in CVD, is maintained and amplified with the participation of both innate and acquired immune responses [23]. Immune cells (macrophages, neutrophils, and lymphocytes) are located in atherosclerotic lesions where proteases are released that degrade the extracellular matrix and cause the rupture of the atherosclerotic plaque. The exposure of all this content with high thrombogenic potential causes the aggregation of platelets and ultimately the formation of the thrombus that obstructs the artery or is released into the circulation [24]. These immune mechanisms interact with acquired metabolic risk factors to initiate, propagate, and activate lesions in the arterial tree [23]. Thus, CVD is now considered a consequence of the immunity/inflammation/thrombosis interaction, key processes in atherothrombosis [25]. Numerous clinical and experimental data support the role of NETs in the initiation and progression of atherosclerotic lesions, inducing and contributing to the formation, stability and growth of arterial thrombus [26]. In murine models, inhibition of NETosis decreased the size of the atherosclerotic lesion and delayed carotid thrombosis [27]. Mice deficient in NE/cathepsin G showed a decrease in the formation of arterial thrombi related to the degradation of TFPI [28]. In humans, Borissoff et al. described for the first time in 2013 an association between elevated levels of surrogate NET markers in plasma [cell free DNA (cfDNA), nucleosomes, and MPO/DNA complexes] and the occurrence of major cardiovascular events [29]. In patients with ST-Elevation Myocardial Infarction (STEMI), these NET components have also been described as biomarkers of cardiovascular events [30]. In addition, they have been located in the atherosclerotic lesion [31,32], positively correlating with the size of the infarct and negatively with DNAse1 activity [32].

In STEMI, NETs at the plaque rupture site contain TF, inducing platelet activation and thrombin generation, which increases their thrombogenic potential [33]. NETs might also facilitate plaque erosion by inducing endoplasmic reticulum stress and endothelial cell apoptosis [34]. In the ASpirin nonresponsiveness and Clopidogrel Endpoint Trial (ASCET) study, with more than 1000 patients with coronary artery disease (CAD), the highest cfDNA levels were associated with twice the risk of suffering an adverse event, regardless of treatment and hypercoagulability markers [35]. Therefore, a central role for NETs in CAD has been proposed. The molecular mechanisms that associate NETosis to the development of atherosclerosis and/or CVD are not fully understood [19]. In the following sections we will describe the role of miRNAs, important modulators of gene expression, in NETosis.

## 3. MicroRNA Biology

The role of microRNAs (miRNAs) as important regulators of gene expression with an impact in disease development was revealed almost twenty years ago [36]. miRNA maturation process is complex [37] and starts in the nucleus with the transcription of the miRNA genes by RNA polymerase II, giving rise to pri-miRNAs that are processed into pre-miRNAs by the microprocessor catalytic complex, composed by DROSHA and DGCR8. Pre-miRNA molecules are 3’ overhang hairpin-like structures of ~60 nucleotides that once formed are exported to the cytoplasm by a RanGTP-dependent exportin (XPO5). In the cytoplasm, pre-miRNAs are further processed by an RNAse III named DICER resulting in the formation of two ~22 nucleotide mature double strand miRNAs that include the mature miRNA guide, -5p, and the complementary passenger strand, named -3p [38]. To allow gene regulation, one or both strands are selected as the guide mature miRNA which is inserted in a functional macromolecular unit that contains among other the argonaute protein 2 (AGO2), where it recognizes its target mRNA, mainly in the 3’ untranslated (3’UTR) region, by partial base pairing in mammalians. The interaction of miRNA with its target provokes mRNA decay and/or protein translation inhibition [39]. To date about 2654 mature miRNAs (Mirbase, Release 22.1: October 2018) have been described in humans. miRNAs may regulate almost all the physiological processes and are involved in the pathophysiology of a number of diseases [40]. Remarkably, miRNAs also target other non-coding RNAs such as long non-coding RNAs (lncRNAs) or circular RNAs (circRNAs) [41]. Indeed RNA-RNA interaction is a novel area whose relevance in cancer has been described [42]. Therefore, it would be wise to explore non-coding RNA networks as molecular mechanisms of NETosis as well as its relevance in pathologies.

## 4. Regulation of NETosis by miRNAs

### 4.1. miR-146a

The critical impact of miR-146a in inflammation is well known and has been extensively documented. This miRNA plays an important role in modulating the inflammatory status through a negative feedback regulating among other the TLR4/NF-κB pathway that promotes the expression of inflammatory cytokines such as IL-6, IL-8, IL-1β, or TNF-α [43]. miR-146a directly regulates the expression of TLR4, Tumor Necrosis Factor Receptor Associated Factor 6 (TRAF6) and Interleukin 1 Receptor Associated Kinase 1 (IRAK1), three essential proteins of the TLR4/NF-κB pathway [44,45,46]. Indeed, the activation of these pathways by pro-inflammatory ligands such as lipopolysaccharide (LPS) also promotes the transcription of miR-146a that may control overwhelmed cellular response to inflammatory signals [44,47,48]. Few years ago, a knock-out mouse model of miR-146a (*miR-146a*^−iR^) unraveled the pathophysiological consequences of miR-146a deficiency [47,49]. As expected, *miR-146a*^−/−^ mice reacted to stresses such as an LPS challenge with an exacerbated inflammatory response generating high serum levels of IL-6 or TNF among others [47,49]. Further, this increased inflammatory response may have consequences in thrombotic diseases as we will describe below. In humans, a partial deficiency of miR-146a exists due to the presence of rs2431697, a miR-SNP located 16 kb upstream of pre-miR-146a. Lofgren et al. showed that the presence of the minor T allele reduces both cellular pri-miR-146a levels as well as mature miR-146a levels (~50%) [50]. Indeed, this natural model of partial miR-146a deficiency has been associated with a series of diseases with a strong inflammatory background. In particular, the presence of this and other miR-SNPs that also decrease the levels of miR-146a, i.e., rs2910164 and rs57095329, are implicated in the development of cardiovascular diseases, cancer or autoimmune diseases [51,52,53]. The first proof that a miRNA can alter NET formation was obtained using a *miR-146a*^−/−^ mouse model and individuals with rs2431697 [54]. Previously, our group demonstrated an association of miR-146a with a higher risk of adverse cardiovascular events (ACEs) in atrial fibrillation (AF) patients bearing the T variant of rs2431697 [55]. We quantified *IL-6* mRNA levels in monocytes carrying rs2431697 CC (*n* = 9) or TT (*n* = 9) genotypes from healthy subjects following the LPS stimulation finding that samples with TT genotype significantly increased *IL-6* expression vs. CC genotype [55]. Although these data helped to explain in part how low levels of miR-146a may affect the development of ACEs in AF patients and by extension the development of thrombotic events in other pathologies, other roles for miR-146a had to be considered. In this sense, a relationship between miRNAs and NETosis was not described at that moment. To investigate a role for miR-146a in NETosis, we quantified the expression of a specific marker of NETosis, citrullinated histone H3 (citH3) and released cfDNA [54]. Activation of *miR-146a*^−/−^ neutrophils with PMA revealed a higher relative increase in cfDNA and citH3-positive cells/total cell ratio in *miR-146a*^−/−^ neutrophils vs. wild-type (WT) littermates. Thus, these results confirmed that miR-146a was involved in NET formation by an unknown mechanism. In order to tie these results to the presence of rs2431697, we recruited healthy donors (TT, *n* = 7 and CC, *n* = 7) and isolated their neutrophils. By performing the same approach, we observed that neutrophils of TT homozygous subjects had higher cfDNA levels and ratio of citH3-positive cells/total cells than CC subjects after PMA activation [54]. Overall, these results showed that a partial/total deficiency of miR-146a modifies the intrinsic capacity of neutrophils to form NETs. Since miR-146a targets potentially involved in this process may be conserved among species, thus the use of *miR-146a*^−/−^ mice as a model of choice to unravel the molecular mechanisms by which this miRNA participate in NETosis is strongly supported. Interestingly, we induced a sterile stress, i.e., atherosclerosis, and a non-sterile, i.e., endotoxemia by LPS [56] in these *miR-146a* deficient mice. Both challenges produced an increased NETosis in *miR-146a*^−/−^ mice vs. WT [56]. In an attempt to better understand how miR-146a exerts its function, the phenotype of *miR-146a*^−/−^ mice was studied. An aging-like phenotype CD62L^low^ CD11b^high^ Cxcr4^high^, with an overexpression of Tlr4 in the aged population was observed [56]. In addition, *miR-146a^−/−^* neutrophils displayed a lower expression of C-X-C Motif Chemokine Receptor 1 (Cxcr1) that has been associated with a pro-inflammatory phenotype [46]. Thus, these results may explain in part how miR-146a deficiency increases NETosis since aged/activated neutrophils have been shown to be more prone to form NETs [57,58]. Additionally, an increased formation of ROS was observed in *miR-146a*^−/−^ neutrophils vs. WT under basal conditions [56]. These latter results suggest that miR-146a deficiency promotes a primed status in neutrophils to form NETs [56]. The molecular mechanisms and miR-146a targets involved in NET formation are still under investigation. How these results obtained in vivo, are transferable to humans is also under investigation, but our results point towards this hypothesis since high levels of NETs due to the presence of rs2431697 are associated with increased ACEs both in atrial fibrillation patients and in community acquired pneumonia patients [54,56]. 

On the other hand, a recent study showed that exosomes derived from macrophages activated with oxidized low-density lipoproteins (ox-LDLs) contained high levels of miR-146a [59]. Authors demonstrated that co-culture of these exosomes with neutrophils promoted the formation of NETs. It was proposed that the transferred miR-146a may down-regulate the expression of superoxide dismutase 2 (SOD2) in neutrophils and thus increase the intracellular levels of ROS which in turn generates NETs. These results suggest that high levels of miR-146a promote NET formation. The difference between these results and ours may reside in the treatment duration. While in our study miR146a deficient neutrophils were incubated with PMA during 4 h, Zhang et al. treated neutrophils with exosomes for 24 h before measuring NETs [59]; how a long time vs. short in culture may affect NETosis needs further evaluation. In addition, it is possible that other factors contained in the exosomes may favor NETosis in Zhang et al. study. Certainly, further investigations are needed to answer many other questions concerning the molecular mechanisms and processes of NETosis regulated by miR-146a. In particular, the role of miR-146a in other cells such as macrophages [60] or platelets [61] that activate NET formation has to be elucidated. Moreover, the discovery of new miR-146a targets involved in NETosis might elucidate its important role in this process. 

### 4.2. miR-155

Another relevant miRNA in inflammation is miR-155 mainly expressed in cells of the hematopoietic system [62]. As it happens for miR-146a, miR-155 expression is up-regulated by several inflammatory stimuli such as TNF, ILs, IFNs signaling through TLRs and NF-κB that directly binds to the promoter of MIR155HG gene (also known as BIC) [63,64]. Interestingly, miR-155 and miR-146a are tightly linked, following an inflammatory stimulus miR-155 is highly transcribed under the control of NF-κB promoting inflammation and with time miR-146a expression increased allowing the attenuation of TLR4/NF-κB pathway which in turn provokes miR-155 down-regulation and inflammation control [64]. Thus, given the relation with miR-146a, miR-155 seems an additional good candidate to regulate NET formation. Hawez et al. showed that miR-155 is also able to regulate NETosis through PAD4 inhibition [65]. In this study, a first result of interest was that NET formation was accompanied by a translational activity [65] which is in contradiction with other previously reported works [66,67]. Indeed, additional studies are needed since this is a critical point concerning the role of miRNAs in NETosis. To know if neutrophil contain all the machinery to allow NET formation or if its activation with agonists triggers the translation of essential proteins for NETosis is very relevant to understand the dynamics of this process and when miRNAs intervene. Their study also showed that, as it has been demonstrated for other miRNAs, the binding of miR-155 to *PAD4* 3′UTR allows a positive activation that increased the levels of *PAD4* mRNA instead of down-regulating the target [65]. In fact, the authors showed increased *PAD4* mRNA levels by transfecting primary human neutrophils with a miR-155 mimic. However, they failed to show this same effect on protein translation in resting and PMA-activated neutrophils. Alternatively, the use of an inhibitor of miR-155 triggered a decrease of both protein and *PAD4* mRNA [65]. The in vitro impact of the deregulation of PAD4 by miR-155 is an elevated NET formation. The relevance of these data has to be investigated in vivo to evaluate the impact of an overexpression of miR-155. For example, it would be of interest to study the impact of the presence of rs767649, a functional miR-SNP that increases the transcriptional activity of the *miR-155* gene [68], on NET formation under basal and pathologic conditions.

### 4.3. miR-505

The next miRNA to be shown as a potential regulator of NET formation was miR-505. This miRNA has been involved in the development of different types of cancer such as breast cancer [69] or pancreatic cancer [70]. Recently, its role in familial hypercholesterolemia has also been characterized showing that miR-505 controls the expression of several chemokine receptors conferring a pro-inflammatory status to patients with this disease [71]. In their work, Chen et al. confirmed that purified exosomes from plasma of patients with atherosclerosis contain higher levels of miR-505 vs. healthy controls [72]. Additionally, they observed that the incubation of HUVECs with ox-LDLs increased the transcription of miR-505 through a NF-κB dependent signal favoring the secretion of exosomes with high miR-505 content [72]. Furthermore, the interaction of these ox-LDL-induced exosomes with human purified neutrophils increased NET formation in vitro [72]. Finally, the authors explained the higher NETosis observed with a down-regulation of NAD-dependent deacetylase sirtuin-3 (SIRT3) by miR-505. SIRT3 is mitochondrial protein that has been implicated in the regulation of ROS production [73]. The authors hypothesized that miR-505 contained in exosomes from atherosclerotic patients are transferred to neutrophils where they downregulate SIRT3 allowing an uncontrolled ROS production that activate NETosis. Interestingly, miR-505 has been tightly linked to high mobility group box 1 protein (HMGB1) expression in different types of cancer [74,75]. How this regulatory pathway may also play a role in NET formation in a cardiovascular context given the essential function of HMGB1 in NETosis may be a point of interest for future studies [76].

### 4.4. miR-378a-3p and miR-15b-5p

Two additional miRNAs were recently described as regulators of NET formation, miR-378a-3p and miR-15b-5p [77]. Authors performed RNA-seq in exosomes purified from platelets activated with LPS or not finding that these two miRNAs were over-expressed in exosomes derived from LPS-treated platelets. Interestingly, these miRNAs potentially regulate phosphatidylinositol 3-kinase PI3K/Akt/mTOR pathway that is actively involved in autophagy [78] that is related with NET formation [79,80]. Incubation of differentiated HL-60 cells with miR-378a-3p and miR-15b-5p mimics showed that they increased NET formation directly regulating the expression of phosphoinositide-dependent kinase-1 (PDK1), an Akt activator. Finally, this work showed that human neutrophils co-cultured with exosomes from LPS-activated platelets internalized more miR-378a-3p and miR-15b-5p than neutrophils incubated with exosomes from non-treated platelets. Overall, these results indicated that platelet derived exosomal miR-15b-5p and miR-378a-3p inhibited Akt/mTOR pathway activity by targeting PDK1 in PMNs, thus promoting NET formation through the activation of autophagy [77].

### 4.5. miR-1696 and miR-16-5p

Finally, two additional studies have been published in a chicken model. The first one described that miR-1696 inhibits glutathione peroxidase 3 (GPx3) and interferes with the formation of ROS and mitogen-activated protein kinase (MAPK) and PI3K/Akt pathways leading to the production of NETs [81]. This study is certainly of interest and further investigation may evidence that GPx3 also plays a role in NETosis in mammals. However, miR-1696 has not been described to date in mammals, but the search for mammalian miRNAs with an effect equivalent to that of chicken miR-1696 on GPx3 and with a subsequent impact on NET formation is guaranteed. 

The second study described the mechanisms explaining the effect of an air pollutant, hydrogen sulfide (H_2_S), on NET formation. The authors revealed that H_2_S up-regulated miR-16-5p expression which impacted in the expression of two potential regulators of NETosis, Raf-1 Proto-Oncogene Serine/Threonine Kinase (RAF1) and Phosphoinositide-3-Kinase Regulatory Subunit 1 (PIK3R1) [82]. In particular, RAF1 is connected to the MEK/ERK/NOX2 pathway and PI3KR1 is associated to the PI3K/Akt/mTOR pathway implicated in autophagy in several cell types and both pathways have been involved in NET formation [83,84]. Therefore, although the study was also performed in a chicken model, there are results of interest that may deserve further investigation in mammals. Specifically, both RAF1 and PIK3R1 are conserved targets of miR-16-5p in mammals and as such variations in this miRNA levels by genetic or environmental factors may have an impact on NETosis [85]. In addition, it would be of interest to investigate if other miRNAs beside miR-16-5p, that also target Raf1 (miR-7-5p [86], miR-370-3p [87], and miR-195/miR-497 [88]) or PIK3R1 (miR-128-3p [89] and miR-486-5p [90]) may regulate NET formation.

## 5. Perspectives

The use of miRNAs as a therapeutic tool against NETosis may also be a potential future application [91]. Specifically, several studies have shown that miR-146a replacement therapy reduces inflammation and thus may benefit thromboinflammatory diseases such as atherosclerosis [92,93]. In the same direction, it has been shown that injection of miR-155 inhibitors reduce the formation of atherosclerotic lesions in mice [94]. Indeed, miR-155 inhibition strategy is already being clinically tested for some hematological diseases such as chronic lymphocytic leukemia (Cobomarsen MRG-106 from Miragen Therapeutics Inc.). To our knowledge, the rest of miRNAs described in this review have not been tested in animal models in the context of cardiovascular diseases, but given their role in NET formation, it would be of interest to investigate it. Thus, the use of exosomes may represent an interesting and yet to evaluate efficient strategy to deliver miRNAs in neutrophils to inhibit NETosis [95]. Approaches to ameliorate the therapeutic efficacy of exosomes or extracellular vesicles (EVs) are necessary. In particular, increasing the loading capacity of mimics or anti-miRNAs in exosomes and constructing exosomes or EVs that express neutrophil-specific antigens to target NET formation are initial essential steps to envision a potential use of miRNAs as therapeutic tools. 

Overall, a better understanding on how miRNA may regulate pathological NETosis is crucial since many questions concerning this relationship remain unanswered (Table 1).

## 6. Conclusions

In this review, we have reported an updated list of the different miRNAs that may be implicated in NET formation (Figure 2). Additional studies will certainly increase the number of miRNAs involved in NETosis. Importantly, miR-146a and miR-155, two endogenous miRNAs with a pivotal role in inflammation, are involved in NET formation although the precise mechanism is still unknown. Whether these miRNAs affect other cells such as platelets that are primordial effectors of NET formation has to be further investigated as well as novel targets within the neutrophil. Remarkably several miRNAs involved in NETosis come from other cells (macrophages, endothelial cells, and platelets) exerting an exocrine effect on neutrophils. It is important to determine if other stresses beside ox-LDL for macrophages and endothelial cells, or LPS for platelets are able to drive the formation of exosomes to carry miRNAs and more importantly, studies have to determine if these exosomes play a role in vivo and to establish in which pathologies this process may occur. Questions remain concerning the amount of miRNAs that has to be transferred to the recipient cells in order to have an effect, a fortiori if these miRNAs are already expressed in neutrophils (e.g., miR-146a and miR-505). Another key point is to know factors that may deregulate the miRNAs expression inducing an inappropriate NET formation. Environmental factors or miR-SNPs affecting miRNAs levels would help to explain the association between the presence of these genetic alterations and thrombosis in patients with diseases with an important inflammatory status.

## Figures and Tables

**Figure 1 ijms-22-02116-f001:**
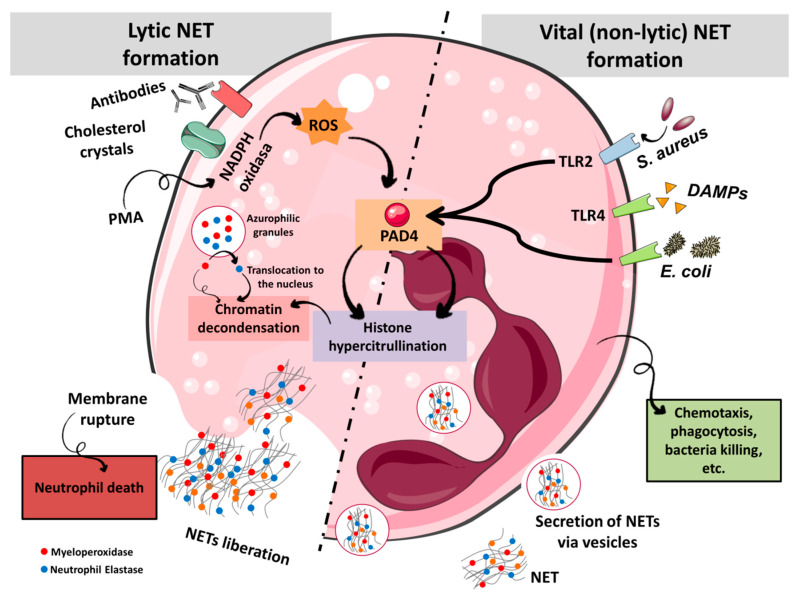
NET formation mechanisms. In response to effectors such as PMA, cholesterol crystals or antibodies, neutrophils perform lytic NETosis. Activation of NADPH oxidase produces ROS that activate PAD4 allowing chromatin decondensation. Concomitantly, neutrophil elastase (NE) and myeloperoxidase (MPO) are translocated into the nucleus and degrade histones, provoking an additional chromatin decondensation. DNA is released into the cytosol and then to the external milieu together with granular and cytosolic proteins inducing neutrophil death. Alternatively, the interaction of neutrophils with bacteria or platelets through TLRs provokes a process called vital NET formation. This mechanism also activates PAD4 and translocation of NE and MPO into the nucleus. Importantly, this process does not provoke the neutrophil death since DNA is secreted via vesicles and instead the cell conserves certain physiological capacities such as phagocytosis.

**Figure 2 ijms-22-02116-f002:**
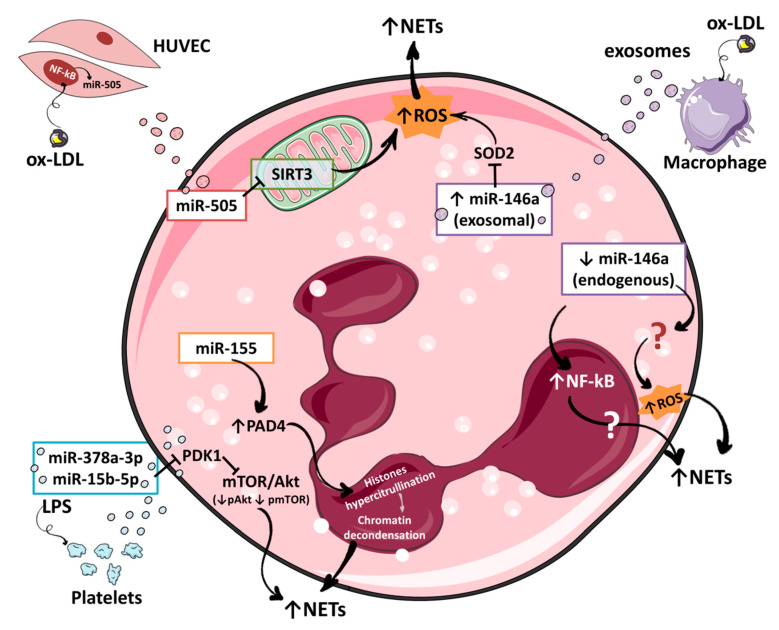
Regulation of NET formation by miRNAs. Endogenous miRNAs such as miR-146a and miR-155 or exogenous miRNAs captured by neutrophils (exosomes) such as miR-505, miR-146a, miR-378-5p or mir-15b-5p regulate several intracellular pathways that inhibit or promote NET formation.

**Table 1 ijms-22-02116-t001:** Issues related to the regulation of NETosis by miRNAs described in this review.

miRNAs	miR-146a	miR-155	miR-505	miR-378a-3p	miR-15b-5p	miR-1696	miR-16-5p
Data in mammalian models	Yes	Yes	Yes	Yes	Yes	No data	No data
Data in CVD	Yes	Yes	Yes	No data	No data	No data	No data
Genetic regulation of miRNA levels	Yes	Yes	No data	No data	No data	No data	No data
Exogenous source	Yes	No data	Yes	Yes	Yes	No data	No data
Mammalian miRNA	Yes	Yes	Yes	Yes	Yes	No data	Yes
Interaction with ncRNAs	No data	No data	No data	No data	No data	No data	No data
Therapeutic potential in CVD	Yes	Yes	No data	No data	No data	No data	No data

CVD: cardiovascular diseases; ncRNAs: non-coding RNAs.

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
