# Peer review of "MicroRNAs as New Regulators of Neutrophil Extracellular Trap Formation"

_ijms, 2021, doi:10.3390/ijms22042116_

Round 1

Reviewer 1 Report

The review by Sonia Aguila et al, describes how miRNAs  regulate neutrophil extracellular trap (NET) formation in disease. The authors discuss interesting aspects of a list of miRNAs known to modulate NET in several inflammatory diseases, specifically in cardiovascular disease. The review offers new insights regarding the role of miRNAs in NET and gives some perspectives for miRNA-based therapies concerning NET-related diseases.

I have some minor points:

1) LINE 56: This sentence is incomplete: “…during and infectious process.”

2) LINE 73: Instead of “several evidences” a more appropriate term to use is: Scientific evidence…(not evidences).

3) LINE 80: punctuation is missing “….important function, such as…”. Also in other parts of the review.

4) LINES 74-80: vital NET and its characteristics are called out in the review. Even if the review calls out a reference n. 9 which describes the types of NET formation,  the function of the two types of NET in the context of the review, should be described. For example, it could be speculated the effect of lytic and non-lytic NET on miRNAs and vice versa: One can think that neutrophils can release miRNAs during the lytic NET, while non-lytic NET retain miRNAs for a determined function. Please explain briefly. An interesting article can be cited here PMID: 32066757.

5) LINE 82: Delete “THUS” and start directly with: “NET formation….”

6) LINE 98: What ST- in STEMI stands for? The same in LINE 112: ASCET: please explain the acronym.

7) LINES 87-88: A brief mention about miRNAs in NET contribution to the activation of coagulation process should be included as point vi) in order to remind that the review is about miRNAs in regulating NETs focusing on cardiovascular disease.

8) More recently published articles should be added:

  1. a) LINE 121 concerning the biogenesis of miRNAs: PMID: 30018188
  2. b) LINE 132: PMID29170536, is an important reference, since the interaction of miRNAs not only with mRNAs but also in general with other non-coding RNAs needs to be considered. MiRNAs make part of molecular networks interactions. This point is very important because gives a perspective of miRNAs interactions with other non-coding RNA species in NETosis, such as lncRNAs applied not only in cancer but also in cardiovascular disease. RNA-RNA networks are an emerging field and a reference should be added here: PMID: 32307915
  3. c) LINE 130: I would rather add the word “about” for the number of mature miRNAs because this number might vary.

9) LINE 205: Rephrase into: ...might elucidate its important role in this process"

10) LINE 280: Please, rephrase this sentence, making it more simple. Also, do the authors mean mammalians models? Such as…?

11) LINE 283: The sentence should be …”may have an impact on NETosis” to be more clear.

12) LINE 277-278: It will be helpful, for the reader, to clarify and recapitulate which are the unanswered questions about the relationship between each miRNA presented here and pathological NETosis. A table with the miRNAs and unanswered questions might be extremely helpful.

13) LINE 312: A reference that describes the use of exosomes as miRNA carriers for therapeutic use: PMID: 25525233 should be mentioned here.

14) I was not able to open the supplementary material given at page 9, line 322. 

Reviewer 2 Report

This is an interesting study merits publication. However, the section on Netosis is immature to publish. To make a good story they need to expand this section especially how NeTosis is playing a role in CAD and cardiovascular diseases is missing. Elaborating detailed mechanisms in atherosclerosis and other cardiovascular diseases will complete this review.

Reviewer 3 Report

Dear authors,

This review attempts to summarize the miRNAs that are or could be related to neutrophil extracellular trap (NET) formation. The manuscript is well structured and written, however some major revisions are necessary, mainly the authors should rewrite the 4.5 section and conclusions. Concretely:

  • Line 46-48: the way in which the authors obtained the information is not of interest. They should remove this sentence.
  • Lines 54, 72, 85, and others: authors' names should not be italicized.
  • Lines 160 and 239: “vs.” should write in italics.
  • Line 167: since the abbreviation WT is the first time it appears in the manuscript; authors should indicate what they refer to.
  • Lines 267-273: Since the relationship between this miRNA and NET is not proven, and this miRNA has not yet been observed in mammals, it should not be included in the review. If, indeed, this miRNA has been related to NET, the authors should expand the information and explain it.
  • Lines 274-283: there are a large number of miRNAs related to the RAF1 and PIK3R1 pathways, although their relationship with NET has not been demonstrated, as is the case with miR-19-5p. Authors should include information about other miRNAs related to this pathway as well.
  • Lines 308-309: This idea has not appeared in the text until now. Authors should include in the miRNA analyzed which references support the claim that any of them could be used as therapeutic.
  • Lines 315-319: This idea is not the subject of the review. This information is not related to the rest of the work, so it should not appear in the conclusions of the same.

Round 2

Reviewer 3 Report

Dear authors,

The manuscript has notably improved compared to the previous version. However, some modifications are still necessary:

  • Line 143: eliminate “the”.
  • Line 152: the first letter of microRNA must be capitalized.
  • Line 153: the authors must indicate the meaning of the abbreviation miRNA, since it is the first time it appears in the text.
  • Line 340: This section should show the conclusions reached by the authors in view of the information provided in the review. For this reason, the references do not make sense in this section, although the information provided is relevant. I recommend to the authors that the information provided (including table 1) is indicated in another section of the review, and in this section, they limit themselves to concluding the relevance of the information provided in the manuscript.

Author Response

Dear authors,

The manuscript has notably improved compared to the previous version. However, some modifications are still necessary:

  • Line 143: eliminate “the”.

Done

  • Line 152: the first letter of microRNA must be capitalized.

Done

  • Line 153: the authors must indicate the meaning of the abbreviation miRNA, since it is the first time it appears in the text.

Done

  • Line 340: This section should show the conclusions reached by the authors in view of the information provided in the review. For this reason, the references do not make sense in this section, although the information provided is relevant. I recommend to the authors that the information provided (including table 1) is indicated in another section of the review, and in this section, they limit themselves to concluding the relevance of the information provided in the manuscript.

We have reorganized the end of the manuscript. We have divided section 5 (“Conclusions and Perspectives”) into two sections. In new section 5 (“Perspectives), we now exclusively describe the use of miRNAs as potential therapeutic tools against NETosis and we include Table 1 in the last sentence of this section.

In section 6 (“Conclusions”), we only summarize the results presented in the Review adding a remark concerning the necessity to investigate other stresses able to promote the formation of exosomes. As suggested by the Reviewer, we now have removed the references from this section.